# Gut Microbiome and Associated Metabolites Following Bariatric Surgery and Comparison to Healthy Controls

**DOI:** 10.3390/microorganisms11051126

**Published:** 2023-04-26

**Authors:** Adilah F. Ahmad, Jose A. Caparrós-Martín, Silvia Lee, Fergal O’Gara, Bu B. Yeap, Daniel J. Green, Mohammed Ballal, Natalie C. Ward, Girish Dwivedi

**Affiliations:** 1Medical School, The University of Western Australia, Perth 6009, Australiamohammed.ballal@uwa.edu.au (M.B.); 2Department of Advanced Clinical and Translational Cardiovascular Imaging, Harry Perkins Institute of Medial Research, Perth 6150, Australia; 3Wal-Yan Respiratory Research Centre, Telethon Kids Institute, Perth 6009, Australia; 4Department of Microbiology, Pathwest Laboratory Medicine, Perth 6000, Australia; 5BIOMERIT Research Centre, School of Microbiology, University College Cork, T12 K8AF Cork, Ireland; 6Department of Endocrinology and Diabetes, Fiona Stanley Hospital, Perth 6150, Australia; 7School of Human Sciences (Exercise and Sport Science), The University of Western Australia, Perth 6009, Australia; 8Department of General Surgery, Fremantle Hospital, Perth 6160, Australia; 9Department of General Surgery, Fiona Stanley Hospital, Perth 6150, Australia; 10Dobney Hypertension Centre, Medical School, The University of Western Australia, Perth 6000, Australia; natalie.ward@uwa.edu.au; 11Department of Cardiology, Fiona Stanley Hospital, Perth 6150, Australia; 12Division of Cardiology, Department of Medicine, University of Ottawa Heart Institute, Ottawa, ON K1Y 4W7, Canada

**Keywords:** obesity, bariatric surgery, cardiovascular disease, dysbiosis, gut microbiome, short chain fatty acids

## Abstract

The gut microbiome plays a significant role in regulating the host’s ability to store fat, which impacts the development of obesity. This observational cohort study recruited obese adult men and women scheduled to undergo sleeve gastrectomy and followed up with them 6 months post-surgery to analyse their microbial taxonomic profiles and associated metabolites in comparison to a healthy control group. There were no significant differences in the gut bacterial diversity between the bariatric patients at baseline and at follow-up or between the bariatric patients and the cohort of healthy controls. However, there were differential abundances in specific bacterial groups between the two cohorts. The bariatric patients were observed to have significant enrichment in *Granulicatella* at baseline and *Streptococcus* and *Actinomyces* at follow-up compared to the healthy controls. Several operational taxonomic units assigned to commensal Clostridia were significantly reduced in the stool of bariatric patients both at baseline and follow-up. When compared to a healthy cohort, the plasma levels of the short chain fatty acid acetate were significantly higher in the bariatric surgery group at baseline. This remained significant when adjusted for age and sex (*p =* 0.013). The levels of soluble CD14 and CD163 were significantly higher (*p* = 0.0432 and *p* = 0.0067, respectively) in the bariatric surgery patients compared to the healthy controls at baseline. The present study demonstrated that there are alterations in the abundance of certain bacterial groups in the gut microbiome of obese patients prior to bariatric surgery compared to healthy individuals, which persist post-sleeve gastrectomy.

## 1. Introduction

The microbiota is a community of microorganisms that consists of commensal, symbiotic, and pathogenic bacteria that live inside the human body. The term microbiome describes the collection of genomes from all the microorganisms. In healthy individuals, it can be classified into 6 bacterial clusters; *Bacillota*, *Bacteroidota*, *Pseudomonadota*, *Actinomycetota*, *Fusobacteriota*, and *Verrucomicrobiota* [1], with more than 90% of the gut microbiota made up of *Bacteroidota* (previously *Bacteroidetes*) and *Bacillota* (previously *Firmicutes*).

Being overweight and obese significantly increases the risk of both type 2 diabetes and cardiovascular disease (CVD) [2]. Bariatric surgery can achieve greater weight loss compared to diet and medication for obesity, especially for morbidly obese patients [3]. There are various types of bariatric surgery including adjustable gastric banding, sleeve gastrectomy, Roux-en-Y gastric bypass (RYGB), and biliopancreatic diversion with a duodenal switch. Observational studies have revealed lower risks of myocardial infarction, stroke, and mortality in bariatric surgery patients, as well as remission or improvement of their diabetes [4,5]. However, randomised controlled trial data contrasting bariatric surgery to intensive nutritional and/or medical therapy are currently lacking [6]. Following bariatric surgery, there is a decrease in skeletal muscle mass, including truncal and appendicular muscle mass [7]. Metabolic improvement also happens rapidly following bariatric surgery, allowing patients with type 2 diabetes to reduce or even stop their glucose-lowering medication [8]. Blood pressure shows a similar prompt response: in hypertensive patients, a drop in blood pressure during the first week of surgery frequently enables a reduction in antihypertensive medication [9]. Endothelial function may also improve; however, there is currently minimal information regarding the timing of any improvements and their relationship to weight loss [10]. Weight loss can improve arterial stiffness, another indicator of vascular function, but the impact of sleeve gastrectomy is still unknown. Similarly, risk of stroke is reduced after bariatric surgery, but the mechanism responsible for this observation is still unknown [11]. Thus, despite the clear metabolic benefits associated with significant weight loss following bariatric surgery, the mechanisms behind these benefits are largely unknown. Alterations in the gut microbiome of obese patients following bariatric surgery have been proposed as a possible mechanism of these benefits [12,13,14]. The hypothesis is based on current research suggesting that alterations in the gut microbiome following bariatric surgery may contribute to the benefits of weight loss. Although the gut microbiome is a complex ecosystem, and although changes in one bacterial species or metabolite may not fully represent the changes occurring in the gut ecosystem as a whole, this study may provide insights into the potential mechanisms underlying the benefits of bariatric surgery and may inform future interventions for obesity and related diseases.

This pilot study sought to investigate the microbiome profiles and associated bacterial metabolites, as well as the markers of inflammation, in a group of obese participants who were at the greatest risk for cardiovascular events before and after gastric sleeve surgery and compare these results to a healthy cohort.

## 2. Materials and Methods

### 2.1. Study Design

This observational cohort study recruited patients scheduled to undergo sleeve gastrectomy at the bariatric surgery unit, St John of God Hospital, Murdoch, Western Australia between August 2019 and July 2021. Inclusion criteria were age 18–59 years and undergoing their first sleeve gastrectomy at the bariatric surgery unit. Exclusion criteria included a history of asthma and kidney problems. All patients provided fasting blood and stool samples at baseline and at 6-month follow-up for assessment of their microbiome, associated bacterial metabolites, and inflammatory profiles. Samples were collected and stored at −80 °C until analysis. A cohort of healthy volunteers were also recruited from the Perth community and provided fasting blood and stool samples for analysis. Healthy participants were defined as individuals with no clinically relevant abnormalities identified by a detailed medical history, a full physical examination including blood pressure and heart rate, and clinical laboratory tests from a fasting blood sample (Figure 1).

### 2.2. Ethical Consideration

Participants were invited to take part in this study during a pre-surgery clinic visit, and written informed consent was obtained prior to inclusion. Participants were provided with a copy of the signed consent forms. This study was approved by the Human Research Ethics committee of St John of God Health Care (SJGHC), Project Reference Number: 1376. Reciprocal approvals were obtained from the University of Western Australia RA/4/20/4731 and Murdoch University 2022/103. The study was conducted in accordance with the Declaration of Helsinki principles and Good Clinical Practice.

### 2.3. Data Collection and Extraction

Detailed medical history, including hypertension, diabetes mellitus, dyslipidaemia, current medical conditions, and medication use was collected from all patients and healthy controls, along with anthropometric and demographic data, including age, height, weight, body mass index (BMI), waist circumference, blood pressure, and heart rate (HR). The 10-year Framingham Risk Score was calculated prior to inclusion in the study [15], and a subset were identified to be at a higher risk of CVD (10-year Framingham Risk Score ≥ 10%). Plasma total cholesterol, high-density lipoprotein cholesterol (HDL-C), low-density lipoprotein cholesterol (LDL-C), triglycerides (TG), and glycated haemoglobin (HbA1c) results were obtained from patient medical records. Systolic blood pressure (SBP) and diastolic blood pressure (DBP) were also measured in clinic.

### 2.4. Microbiome Profile

For analysis, the stool was thawed, stirred well to homogenise, and 80 mg was transferred into screw-top lids for DNA extraction. The shape and texture of the samples were rated using the Bristol Stool Chart. Any possible pathogens present in the stool were inactivated. This was performed as a preprocessing step and involved adding beads (0.1 g of zirconia/silica beads (0.1 mm diameter, Daintree Scientific, Tasmania, Australia) to the stool samples with 500 µL of lysis buffer and 50 µL proteinase K and vortexing well. The samples were then bead beaten using a thermomixer (4 × 1 min pulses set at 50 oscillations). Once bead beaten, the samples were heated to 56 °C for 1 h, vortexed, and again heated at 95 °C for 10 min to inactivate any pathogens. Once this was completed, 300 µL of CD2 was added from the PowerSoil kit (QIAGEN DNeasy 96 PowerSoil Pro QIAcube HT Kit, Cat. No./ID 47021, QIAGEN, Hilden, Germany) and 550 µL was passed into a 96-well S-Block for DNA extraction. For DNA extraction, 550 µL of CD3 solution was added to the samples in the S-blocks and vortexed. Using a vacuum, a QIAamp 96 plate was used to bind DNA. The block then underwent a series of centrifuging and washing processes using 500 µL of Buffer AW1 and then AW2 before a final wash with ethanol. DNA was then eluted with solution C6. Concentration of DNA was evaluated using a NanoDrop spectrophotometer with a 1–2 µL sample. The gut microbiota was profiled following an amplicon sequencing approach targeting the V3–V4 region of the 16S rRNA gene. Libraries were generated using the MiSeq Reagent Kit v2, and 300 bp reads from paired-end configuration were conducted and sequenced in a MiSeq instrument at the Ramaciotti Centre for Genomics (UNSW, Sydney, Australia).

### 2.5. Short Chain Fatty Acid Assay

Plasma short chain fatty acid (SCFA) concentrations were analysed using gas chromatography mass spectrometry (GC-MS), as previously described, with 10mM each of sodium acetate (^13^C-Acetate, 99 atom % ^13^C, Sigma-Aldrich, St. Louis, MO, USA), sodium propionate (^13^C-Proprionate, 99 atom % ^13^C, Sigma-Aldrich), and sodium butyrate (^13^C-Butyrate, 99 atom % ^13^C, Sigma-Aldrich), as internal standards [16]. Briefly, 10 µL of 0.5M HCl and 20 µL of internal stock standard solution was added to 100 µL of thawed plasma. The samples were vortexed before addition of 300 µL of isopropanol (2-Propanol, ThermoFisher Scientific, Waltham, MA, USA, 99%) for homogenization. Samples were then vortexed and centrifuged, after which 80 µL of supernatant was run on the GC-MS (Agilent HP 6890 Series GC System, 5973 Mass Selective Detector, 7683 Series Autosampler and 7683 Series Injector, Santa Clara, CA, USA) with a DB-Waxetr column (30 cm length, 0.25 mm diameter, and 0.25 µm film), with helium as the carrier gas, according to previously published specifications [16]. Sample concentrations of SCFAs were calculated by monitoring the following ions: 60 and 61 *m*/*z* (acetate), 74 and 75 *m*/*z* (propionate), and 60 and 61 *m*/*z* (butyrate). The retention times were determined from an internal standard. Sample concentrations of SCFAs were calculated by comparing the peak area for each SCFA against the labelled internal standard.

### 2.6. CD14 and CD163 ELISA

Commercial antibody pairs were used to assess serum levels of soluble CD14 (sCD14) and soluble CD163 (sCD163) (R & D Systems, Minneapolis, MN, USA) in accordance with the manufacturer’s instructions. Briefly, half volume ninety-six well plates (Corning, Corning, NY, USA) were coated with the antibodies to the soluble marker of interest and incubated overnight at 4 °C. Plates were washed with 0.05% Tween/PBS and blocked with 1% BSA/PBS for 1 h to prevent non-specific binding. Standards and test sera were added to the plates and incubated for 2 h. Plates were washed and incubated with biotinylated antibody. After 2 h, plates were washed and incubated with Streptavidin-HRP for 20 min. Plates were washed again, TMB substrate was added, and the plates were left in the dark for colour development. The enzyme was inactivated via addition of H_2_SO_4_, and the absorbance was read on the spectrophotometer at 450 nm. The coefficients of variance for each assay were sCD14 (6.5%) and sCD163 (10.5%).

### 2.7. Statistical Analysis

All calculations were conducted using SPSS (Statistical Package for the Social Sciences, IBM SPSS Statistics for Mac, version 28 (IBM Corp., Armonk, NY, USA)) and GraphPad Prism software (Version 9.0 for MacOS, GraphPad Software, San Diego, CA, USA). Results are presented as mean ± standard error of mean (SEM) or medians and interquartile ranges as appropriate. Differences within groups at baseline and follow-up were analysed using a paired *t*-test. Differences between groups at baseline were analysed using an independent *t*-test. Differences between groups at follow-up were analysed following correction for baseline using a univariate general linear model with adjustments for multiple comparisons. For overall group comparisons a Mann–Whitney test was used. Matched analyses were repeated using Wilcoxon matched pairs signed rank tests. Correlations were analysed by Spearman’s rank tests. Results were adjusted for age, sex, and BMI. A two-tailed *p* value < 0.05 was considered statistically significant.

#### Microbiome Sequencing Analysis

Read quality assessment and preprocessing of raw sequences was conducted as described [17]. Joint reads were then analysed following the SILVAngs pipeline and taxonomically annotated using the standardized SILVA SSU taxonomy (release 138.1) [18,19]. Taxonomic profiles were analysed in R (Version 4.0.2) using built-in functions for statistical assessment, the R package vegan [20] for estimating parameters of microbial ecology, and the ANCOM-BC method for differential abundance analysis [21].

## 3. Results

### 3.1. Patient Characteristics

Seventeen patients, 6 (35.3%) male and 11 (64.7%) female, were included in the study, and their baseline characteristics are presented in Table 1. The average age was 51.5 ± 7.8 years, the average weight was 127.5 ± 30.2 kg, and the average BMI was 41.4 ± 9.7 kg/m^2^. Eighty-two percent of patients did not smoke, one (5.9%) was a former smoker, and two (11.8%) were current smokers. Six patients had a Framingham Risk Score ≥ 10% at baseline. Four (23.5%) patients had type 2 diabetes, with two of those patients receiving diabetic treatment. Nine (52.9%) patients were hypertensive, and all were receiving antihypertensive medication. Six (35.3%) patients were receiving statin therapy. At baseline, glucose levels were 5.8 ± 1.3 mmol/L, total cholesterol levels were 4.6 ± 1.1 mmol/L, LDL-C levels were 2.6 ± 0.3 mmol/L, and triglyceride levels were 1.6 ± 0.6 mmol/L. Thirteen patients returned at 6 months follow-up. At follow-up, the average weight was 89.1 ± 23 kg, and the average BMI was 31.2 ± 7.4 kg/m^2^ (Table 1). Furthermore, a reduction was seen in both fasting glucose levels and lipid profiles (Table 1). At baseline, 59 healthy volunteers were recruited into the study, and 53 (89.8%) came back for the 6-month follow-up. The baseline and follow-up characteristics of healthy controls are presented in Appendix A.

### 3.2. Microbiome Profile

There were no significant differences in alpha diversity in the gut microbiota of bariatric patients at baseline versus follow-up (Wilcoxon Rank Sum Test (WRST), *p* = 0.94). Similar results were obtained when the analysis was performed on paired samples of patients who provided both baseline and follow-up data (paired WRST, *p* = 0.57) (Figure 2A,B). Similarly, the differential abundance analysis did not identify significant features between time points (Figure 2C). There were no significant differences in alpha diversity in the gut microbiota of healthy volunteers at baseline versus follow-up as a whole cohort (WRST, *p* = 0.84) or as paired analysis (paired WRST, *p* = 0.61) (Appendix A). The differential abundance analysis did not identify significant features between time points.

We did not observe differences in alpha diversity in stool between the bariatric patients and the healthy controls either at baseline (WRST, *p* = 0.30) or follow-up (WRST, *p =* 0.20) (Figure 3A,B). Conversely, the bariatric patients showed marked differences in the stool taxonomic profiles at baseline and at follow-up when compared to the healthy controls. Thus, 14 taxonomic features were found significantly enriched and 1 was found significantly depleted in the bariatric patients compared to the healthy controls at baseline (Figure 3C). Similarly, the bariatric patients attending the 6-month follow-up visit had stool-associated bacterial profiles different to those of the healthy controls (Figure 4). Operational taxonomic units (OTUs) assigned to taxa that are typically associated with the oronasal cavity, such as *Granulicatella* (baseline, Figure 3C) or *Streptococcus* and *Actinomyces* (follow-up, Figure 4) [22], were significantly enriched in patients undergoing bariatric surgery. Conversely, several OTUs assigned to commensal Clostridia were significantly reduced in the stool of bariatric patients both at baseline and follow-up (Figure 3C and Figure 4).

### 3.3. Short Chain Fatty Acids

Within the bariatric surgery group, among those who provided blood samples at both baseline and follow-up, plasma SCFA levels were not significantly different from baseline to 6 months post-surgery. Plasma levels of acetate, propionate, and butyrate in the patients pre- and post-bariatric surgery and in the healthy control group are shown in Figure 5. At baseline, levels of acetate in the bariatric patients were significantly elevated compared to the healthy group (*p* = 0.003). When adjusted for age and sex, this remained significant (*p* = 0.013). Propionate and butyrate were not significantly different between the bariatric patients and the healthy controls before or after adjusting for their baseline levels and for age and sex.

### 3.4. CD14 and CD163 Inflammatory Profile

Serum levels of soluble CD14 and CD163 were analysed in the bariatric and healthy groups as markers of monocyte/macrophage activation and inflammation. The levels of sCD14 were not significantly different within the overall and paired bariatric patients at baseline compared to follow-up. sCD163 levels were significantly higher in the overall bariatric patients at baseline compared to follow-up at *p* = 0.049; however, when only paired samples were analysed, this was no longer significant. At baseline, sCD14 and sCD163 levels were significantly higher in the bariatric patients compared to in the healthy controls at *p* = 0.043 and *p* = 0.007, respectively. sCD14 and sCD163 remained significant following the adjustment for age and sex (*p* = 0.005 and *p* < 0.001, respectively). The levels of sCD14 and sCD163 were not significant at the 6-month follow-up in the bariatric patients compared to the healthy controls (Figure 6). In the bariatric cohort, the levels of sCD14 did not correlate with the levels of sCD163 at baseline (r = −0.2, *p* = 0.613) or follow-up (r = 0.4, *p* = 0.419). In the healthy cohort, sCD14 levels did not correlate with the sCD163 levels at baseline; however, at follow-up, there was a significant positive association between sCD14 and sCD163 (r = 0.4, *p* = 0.003).

## 4. Discussion

The mechanisms underlying clinical improvements, including lipid profiles and glycaemic control, observed after bariatric surgery are not fully understood, although it has been postulated to be linked in part to body composition changes [23,24,25]. This study sought to determine if these improvements are associated with changes in the gut microbiome profile and associated metabolites, as well as with inflammation in obese patients undergoing sleeve gastrectomy surgery. Additional comparisons with a healthy control group were also made. We did not observe significant changes in gut bacterial diversity in the bariatric patients pre- or post-surgery nor between the bariatric patients and the healthy cohort. We did, however, see marked differences in the stool taxonomic profiles and a significant increase in circulating acetate levels and in the markers of inflammation in pre-surgery obese patients compared to the healthy controls.

The gut microbiome has been extensively studied in the context of obesity and cardiovascular disease [26]. In this study, we evaluated parameters of gut ecology in patients undergoing bariatric surgery and healthy controls both at baseline and at the follow-up visit. Patients undergoing bariatric surgery were characterized by a severe reduction in several members of commensal Clostridia, which, apart from being SCFAs producers, are also involved in the maintenance of homeostasis in the gastrointestinal tract [27]. These profiles agree with our results, showing a reduction in acetic acid and increased inflammation prior to surgery. Conversely, the bariatric patients showed a higher abundance of microorganisms typically associated with the upper aerodigestive system such as *Granulicatella*, *Streptococcus*, or *Actinomyces*, even at 6 months post-surgery. These taxonomical entities have been previously linked to obesity [25] and suggest changes in the physicochemical parameters of the gastrointestinal environment. Gut alpha diversity was comparable between visits (baseline versus follow-up) and conditions (bariatric surgery versus the healthy controls), suggesting that neither the disease state nor the time of sampling had a significant impact on the overall structure of the gut bacterial communities. While previous studies have observed taxonomic changes in the gut microbiota of obese patients after undergoing bariatric surgery [28], not all patients show increases in gut bacterial diversity associated with improvements in metabolic phenotype. A previous study in 61 subjects with severe obesity undergoing gastric banding or Roux-en-Y-gastric bypass who were followed for up to 12 months concluded that 75% of patients had a low gut microbial richness, which correlates with the metabolic parameters present prior to surgery [12]. Interestingly, in most Roux-en-Y-gastric bypass patients, gut bacterial diversity remained unaltered even after a one-year follow-up, despite improvement in metabolic parameters [12]. Sleeve gastrectomy involves removing most of the stomach in a vertical direction, leaving a tube-shaped section on the side of the stomach. Meanwhile, in Roux-en-Y gastric bypass, a small pouch is created from the stomach and connected to the small intestine, bypassing the upper part of the digestive tract [29]. While it is true that we observed that several OTUs were differentially regulated between the bariatric patients and the healthy controls, the interindividual variability was high, as suggested by the spread in the confidence intervals. It is important to note that our cohort underwent sleeve gastrectomy, and due to the small sample size, the ability to compare between different procedures is limited.

Acetate was significantly elevated in the bariatric patients when compared to the healthy controls at baseline. Acetate is involved in the production of energy, the synthesis of lipids, and the acetylation of proteins [30]. This is in line with our obese cohort having a significantly larger BMI, thus requiring more energy production and lipid synthesis. Some animal studies have suggested a role for acetate in the development of obesity and metabolic syndrome [31]. In contrast, other animal studies have shown the role of acetate in reducing appetite by directly affecting the hypothalamus [32], blocking endogenous lipolysis [33], increasing hepatic absorption of blood cholesterol [34], and reducing hyperglycaemia [35], and others have suggested it is associated with greater gut microbiome diversity and lower visceral fat [36], with acetate supplementation able to suppress weight gain in high-fat-fed mice [37,38]. Human studies, however, are inconsistent [39,40], further highlighting a need to investigate the metabolic effects of acetate.

There is a strong link between inflammation and obesity. Obesity is associated with chronic low-grade inflammation, which contributes to the development of obesity-related diseases, such as type 2 diabetes and CVD. In our study, the bariatric patients had significantly elevated sCD14 levels compared to the healthy controls, suggesting an increased inflammatory state. sCD14 levels have been linked to obesity-induced vascular impairment [41] and are also associated indirectly with the gut microbiome via bacterial-derived lipopolysaccharide (LPS), which plays a role in the inflammatory process and the onset of obesity-related illnesses [42,43]. Research suggests that an altered gut microbiota and LPS translocation may be early drivers of insulin resistance, diabetes, and obesity [44]. An eight-week overfeeding period in non-obese people raised LPS and the sCD14 ratio in a fasting state, as well as the postprandial accumulation of LPS [45]. In a subsequent study of obese patients who received RYGB surgery compared to nonobese patients with no comorbidities, the obese group had significantly greater sCD14 expression on monocytes [44,46] compared to the controls, which then normalized from 6 to 12 months post-surgery [46]. This is in line with our results showing elevated levels of sCD14 in the bariatric patients compared to the healthy controls, which normalized 6 months post-surgery to levels seen in the heathy controls alone, suggesting an increased state of inflammation, which may be associated with gut microbiome dysbiosis. sCD163 is a macrophage marker with anti-inflammatory effects, with its soluble form (sCD163) thought to be a predictor of chronic disease including type 2 diabetes and obesity [47,48,49,50] as well as inflammation [51,52,53]. In the present study, we found increased sCD163 levels in bariatric patients compared to the healthy cohort at baseline, which normalized 6 months after surgery to levels seen in the healthy controls.

It is important to note that several limitations exist in the present study, including the small number of patients, particularly in the bariatric surgery group at the 6-month follow-up. This may have limited our ability to detect significant changes in gut microbiome composition, metabolite levels, and inflammation, particularly considering the significant role of other factors, including diet, medication, and other comorbidities. Furthermore, the relatively short follow-up period of 6 months may not be sufficient to fully capture the changes that occur in the gut microbiome after bariatric surgery. Differences in methodology between our study and others, including in the measurement of gut metabolites and the type of bariatric surgery, may also have contributed. Additionally, the observational nature of the study meant that laboratory and clinical data collected from patient notes was not sufficient to determine fasting status, medication adherence, or the completeness of medical history. Dietary and lifestyle information were also not collected, both of which are known to impact the microbiome, particularly after bariatric surgery, when diet and portion sizes are likely to have changed. Similarly, the study did not investigate the potential impact of physical activity or exercise on weight loss or microbiome changes. Finally, the study only included patients from a single centre, which may limit the generalizability of the findings. It would also be an added benefit to incorporate more predicted data analysis trying to link microbiota and metabolites virtually or incorporating more metabolomics and metaproteomics. Future research that considers larger cohorts and longer follow-up periods and that adjusts for energy and food intake, as well as the rate of weight reduction post-bariatric surgery, is required to further establish the contribution of the gut microbiota to weight loss and inflammatory profiles after bariatric surgery.

## 5. Conclusions

In summary, the present study has demonstrated that there are alterations in the abundance of certain bacterial groups in the microbiome of bariatric patients prior to surgery compared to healthy individuals, which persist following sleeve gastrectomy. In addition, there are differences in the levels of acetate and sCD14 and sCD163 between pre-surgery bariatric patients and healthy individuals, which suggests an imbalance in energy homeostasis and inflammation. Further investigations on the inter-relationship between microbiome composition and systemic metabolic and inflammatory processes in the bariatric setting are warranted.

## Figures and Tables

**Figure 1 microorganisms-11-01126-f001:**
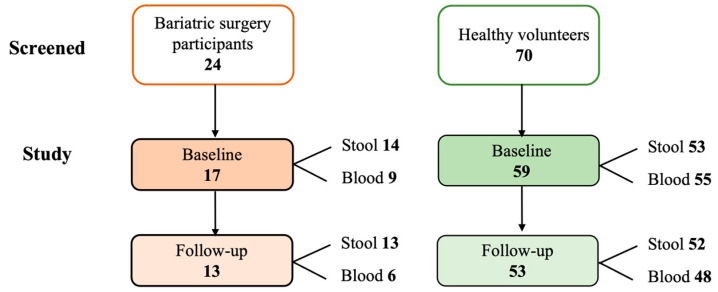
Participant flow in the study.

**Figure 2 microorganisms-11-01126-f002:**
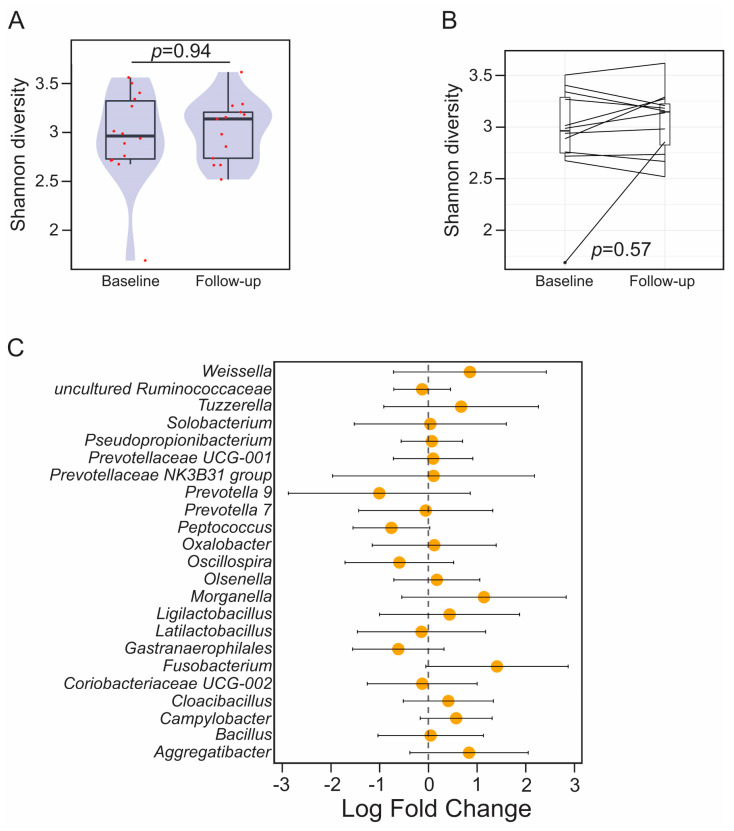
Analysis of microbiome diversity and composition in bariatric patients at baseline and 6 months follow-up. (**A**) Barplots overlaid with density plots showing Shannon diversity at baseline and follow-up for all bariatric patients. Red dots represent the diversity value for each sample analysed. (**B**) Barplots showing the alpha diversity (Shannon index) for the paired data obtained at baseline (left) and follow-up (right). Samples collected from the same patient are connected with a solid line. (**C**) Forest plots representing the coefficients from the ANCOM-BC model with 95% confidence interval. Features enriched at baseline and follow-up are represented with negative and positive fold change values, respectively. Each OTU is represented at the lowest taxonomic rank at which it could be classified according to SILVA taxonomy (release 138.1).

**Figure 3 microorganisms-11-01126-f003:**
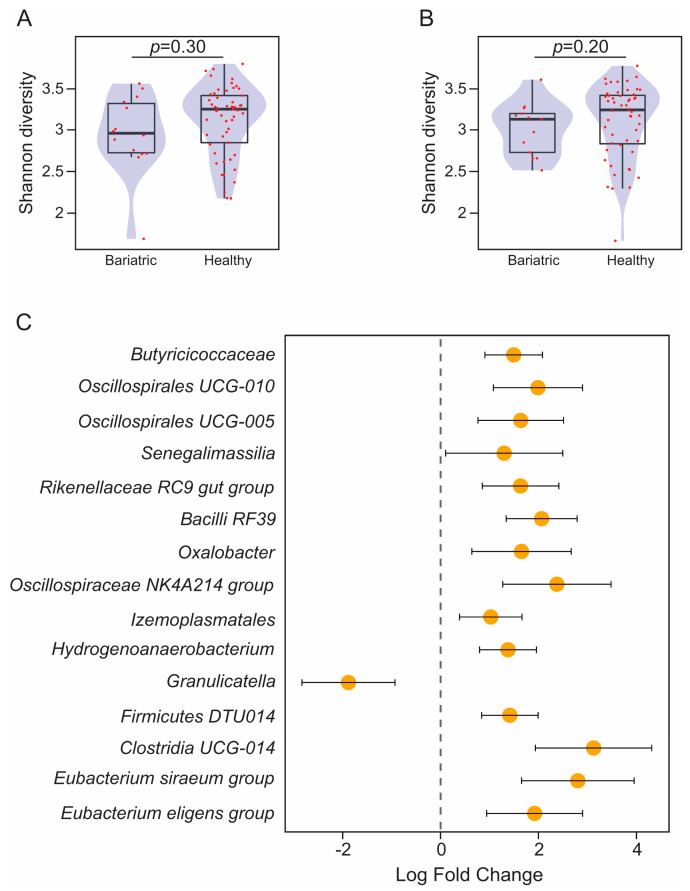
Barplots overlaid with density plots showing gut microbiome ecological diversity determined by Shannon diversity in bariatric patients compared to healthy controls at (**A**) baseline and (**B**) 6-month follow-up. (**C**) Forest plots representing the coefficients from the ANCOM-BC model with pointwise 95% confidence interval. Features enriched in bariatric patients and healthy controls are represented with negative and positive fold change values, respectively. Only features with a fold change higher than |1| are shown. Each OTU is represented at the lowest taxonomic rank at which it could be classified according to SILVA taxonomy (release 138.1).

**Figure 4 microorganisms-11-01126-f004:**
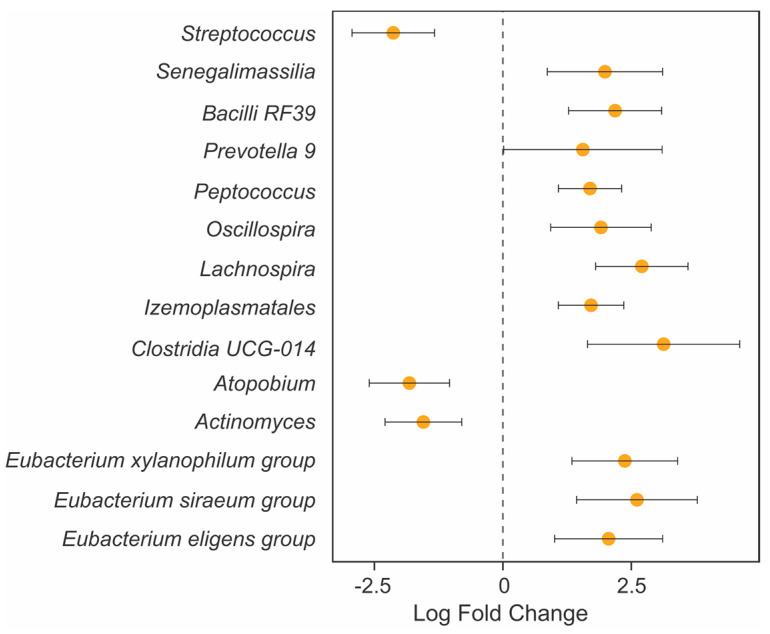
Forest plots representing the coefficients from the ANCOM-BC model with pointwise 95% confidence interval. Features enriched in bariatric patients and healthy controls are represented with negative and positive fold change values, respectively. Only features with a fold change higher than |1.5| are shown. Each OTU is represented at the lowest taxonomic rank at which it could be classified according to SILVA taxonomy (release 138.1).

**Figure 5 microorganisms-11-01126-f005:**
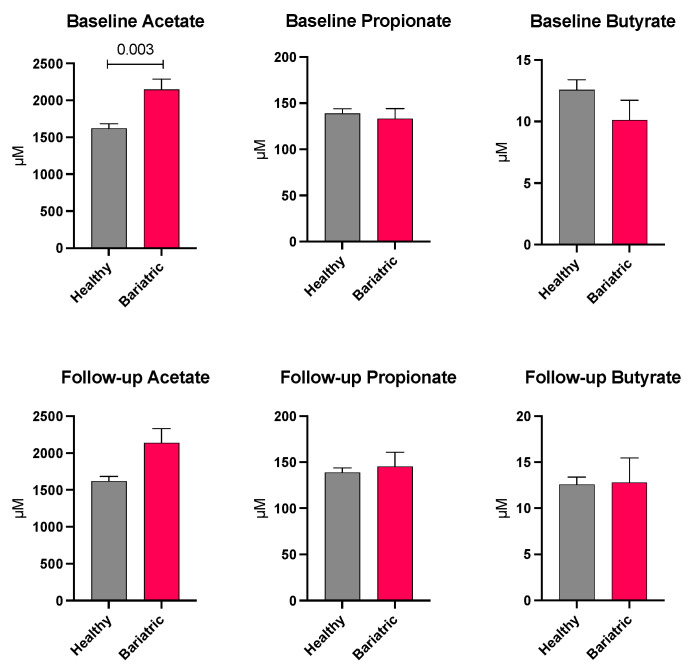
Unadjusted plasma acetate, propionate, and butyrate levels in healthy and bariatric groups at baseline and 6-month follow-up.

**Figure 6 microorganisms-11-01126-f006:**
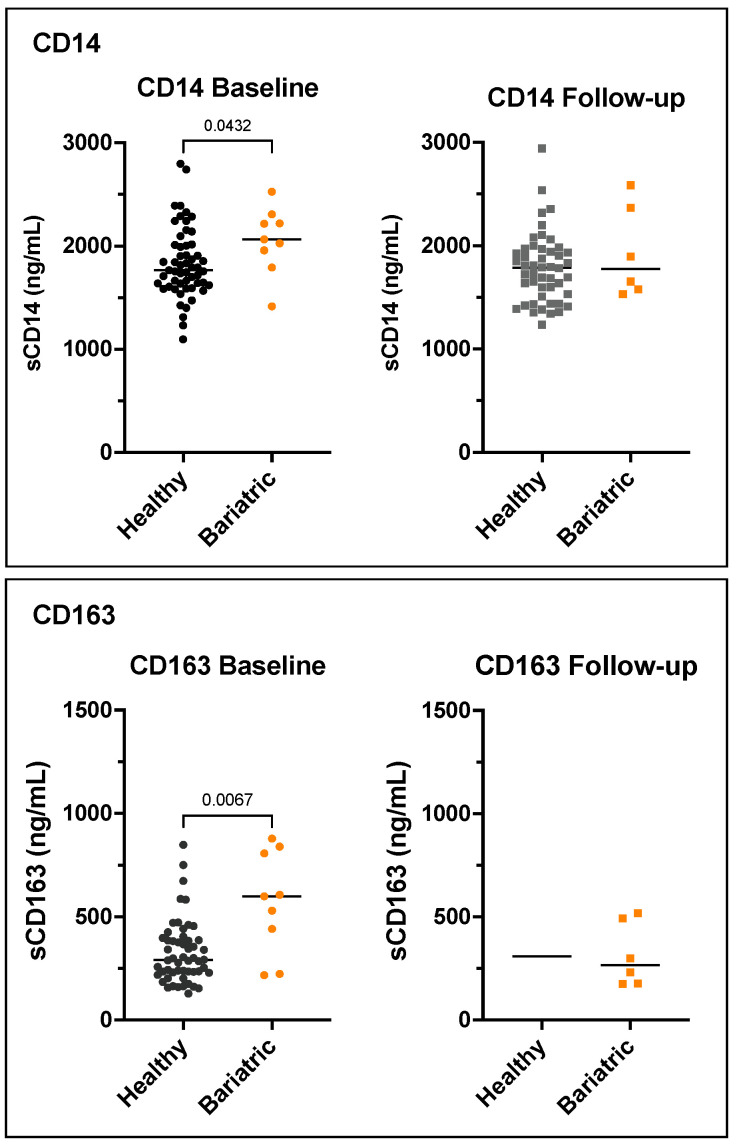
Unadjusted serum sCD14 (**top**) and sCD163 levels (**bottom**) in bariatric group compared to healthy controls. Both were significantly elevated in the bariatric cohort prior to sleeve gastrectomy surgery. Horizontal line represents the median.

**Table 1 microorganisms-11-01126-t001:** Baseline and follow-up characteristics of patients who underwent bariatric surgery.

Bariatric Surgery Patients			
	Pre-Surgery	6 Months Post-Surgery	*p* Value
n	17	13	
Age, years	51.5 ± 7.8	50.9 ± 8.2	0.84
Weight, kg	117 ± 30.2	89.1 ± 23.0	0.01
BMI, kg/m^2^	41.4 ± 9.7	31.2 ± 7.4	0.004
Waist circumference, cm	127.5 ± 17.9	98.5 ± 14.2	<0.001
Men, n (%)	6 (35.3)	3 (25)	0.083
SBP, mmHg	125 ± 12.8	114 ± 9.9	0.016
DBP, mmHg	75 ± 8.1	67 ± 9.7	0.020
HR, bpm	70 ± 9.6	62 ± 10.3	0.037
Laboratory			
Glucose, mmol/L	5.8 ± 1.3	4.9 ± 0.6	0.029
HbA1c, %	6.0 ± 0.7	5.3 ± 0.5	0.005
Total cholesterol, mmol/L	4.6 ± 1.1	4.3 ± 0.7	0.398
LDL-C, mmol/L	2.6 ± 1.0	2.3 ± 0.8	0.383
HDL-C, mmol/L	1.2 ± 0.3	1.4 ± 0.2	0.047
Triglycerides, mmol/L	1.6 ± 0.6	1.2 ± 0.4	0.047
Smoker, n (%)			
Yes	2 (11.8)	-	-
No	14 (82.4)	-	-
Ex	1 (5.9)	-	-

Values are mean ± SD, n, n (%). Significant.

## Data Availability

Data can be made available upon reasonable request to the corresponding author if permitted by the relevant ethics committee.

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
