# Peer review of "Gut Microbiome and Associated Metabolites Following Bariatric Surgery and Comparison to Healthy Controls"

_microorganisms, 2023, doi:10.3390/microorganisms11051126_

Round 1
Reviewer 1 Report
The authors in this study analysed the microbiota by 16S rRNA gene profiling in a small cohort of patients before and after gastric-sleeve surgery and compared them to a healthy control. A number of bacteria taxa were identified at different relative abundances in patients before and after surgery. Authors identified Acetate as the only metabolite which was significantly altered between patients before surgery compared to healthy controls. In the follow up analysis after surgery no significant changes in SCFA concentrations were observed. Furthermore, two inflammation markers were measured in blood sampled, namely CD14 and CD 163 These did not change significantly between baseline and follow-up after surgery. Though, a significant difference was observed between the healthy controls and surgery group at the baseline.
Though the authors had a very interesting cohort, the analyses done could have been greatly improved, by analysing more features such a wider range of metabolites. The findings are very descriptive. The manuscripts could have further been improved by better linking the microbiota and host findings. For a manuscript submitted to Microorganisms it contained only scarce analysis on the microbiota.
Major points:
1. Why were short chain fatty acids not measured in stool this could have given hints if the changes observed in blood acetate levels were linked to changes in the microbiota.
2. Only doing 16S rRNA analysis on the microbiota is not sufficient to characterize and analyse changes in the microbiota. A more functional approach would have been better such as metabolomics, metaproteomics or a functional prediction of the 16S rRNA gene data.
3. Blood analysis was only based on three SCFAs and two inflammation markers.
4. A more comprehensive metabolic analysis should have been done on blood as well as stool to improve study and thereby also being able to link changes in the microbiota with those identified in the host.
Minor points:
1. In the abstract the authors summarize that “The present study demonstrated that there are alterations in the abundance of certain bacterial groups in the gut microbiome of bariatric patients prior to surgery compared to controls, which persist post- sleeve gastrectomy”. Should it not read obese patients (or similar) compared to controls, which persist post- sleeve gastrectomy.
2. The changes mentioned in the abstract did not persist between baseline comparison and follow-up comparison of patients and control. The different relative abundant taxa at baseline were not the same as those identified at follow-up.
3. In the caption of figure 2C it reads: Features enriched at baseline and follow-up are represented with negative and positive fold change values respectively. Would it not make more sense to have enrichment in baseline be negative while enrichment in follow up positive fold change values?
4. The figures representing the data of the taxonomic analysis (figures 3 and 4) show taxa on multiple taxonomic levels, although most are on genus level. The authors should either choose a specific level for the analysis or have a separate figure for the levels discussed
5. Figure 5 the bar charts for the follow-up groups are not coloured
6. In the discussion the authors compare their findings to other bariatric surgery studies. Including Roux-Y-Gastric-Bypass studies. Most readers of microorganism will not know the differences in the gut architectures and environment following RYGB and gastric-sleeve. This should be briefly explained which would explain maybe differences in findings between gastric-sleeve and RYGB studies.
Author Response
Reviewer 1:
1. Why were short chain fatty acids not measured in stool this could have given hints if the changes observed in blood acetate levels were linked to changes in the microbiota.
While both plasma and faecal short chain fatty acids (SCFA) provide important information regarding gut and overall health, they represent different aspects of SCFA metabolism in the body. The SCFA found in stool are SCFA that have not been absorbed and represent the levels excreted, whilst plasma SCFA have been absorbed from the gut and reflect circulating levels, which have the ability to be taken up and act in tissues. They can be used as an energy source by various tissues and organs, and they have been linked to improvements in insulin sensitivity and glucose metabolism. In the present study, we were interested in correlating gut microbiota composition with circulating levels of plasma SCFA.
2. Only doing 16S rRNA analysis on the microbiota is not sufficient to characterize and analyse changes in the microbiota. A more functional approach would have been better such as metabolomics, metaproteomics or a functional prediction of the 16S rRNA gene data.
16S rRNA technology analysis is the currently recognized “Gold Standard” methodology to analyse microbiota in biological systems. We have used this approach appropriately and to internationally accepted standards in our study to address the issue of microbiota diversity and abundance in the bariatric cohort under investigation. An important goal for many human microbiome studies is to identify the metabolic functions of microbes in complex microbiota communities and their impact on host physiology, however such research approaches can be confounded by inadequately understood effects of community composition and host factors on the metabolic traits of specific taxa. For this reason, a prime objective of this investigation was to focus on characterizing the microbiota. Our work now underpins the next phase of investigation, which will be aimed at deciphering and understanding how microbe ecological interactions and metabolite diversity in the gut influence host traits.
In the absence of detailed genomic information on the nature of the microbiota composition, application of “omics” approaches such as metabolomics and metaproteomics were not appropriate functional approaches that could have been applied in our sample analysis. Furthermore, a virtual functional prediction of the 16SrRNA marker gene data was not included in the manuscript because of recognized limitations with such an approach. These limitations range from difficulties with lack of microbial gene annotation accuracy, lack of reference genomes and functional redundancy associated with complex microbial communities.
These recognized limitations with functional predictions can make it difficult to make relevant biological interpretations related to microbiota and metabolites in our bariatric patients. As pointed out by Reviewer 2, a strength of the Manuscript centres on the fact that the data sets included for microbiota and metabolites are all experimentally generated. Consequently, a virtually predicted approach is not included in the Manuscript. However, the limitations of our approach are now further addressed in the Manuscript, Page 11, Lines 400 – 405.
3. Blood analysis was only based on three SCFAs and two inflammation markers.
We acknowledge that our blood analysis was limited to only three SCFAs and two inflammation markers. Our pilot study aimed to investigate the potential relationship between obese patients undergoing bariatric surgery and the gut microbiota and we chose these biomarkers as they are more relevant to human health based on their already established roles in the literature regarding human microbiome studies. While we agree that a more comprehensive analysis of blood biomarkers could provide a more thorough understanding of the potential impact of the microbiota, as this was a pilot study further analysis of additional metabolites was beyond the scope of the study.
4. A more comprehensive metabolic analysis should have been done on blood as well as stool to improve study and thereby also being able to link changes in the microbiota with those identified in the host.
We acknowledge that additional analysis could have been performed on blood in addition to the stool, however, as outlined in our response above (1 and 2), more comprehensive analysis was beyond the scope of this pilot-study.
5. In the abstract the authors summarize that “The present study demonstrated that there are alterations in the abundance of certain bacterial groups in the gut microbiome of bariatric patients prior to surgery compared to controls, which persist post- sleeve gastrectomy”. Should it not read obese patients (or similar) compared to controls, which persist post- sleeve gastrectomy.
Thank you for this suggestion. We have amended the wording of the sentence for improved readability.
6. The changes mentioned in the abstract did not persist between baseline comparison and follow-up comparison of patients and control. The different relative abundant taxa at baseline were not the same as those identified at follow-up.
We have amended the abstract to further clarify which bacterial groups were different at baseline and at follow-up (line 33).
7. In the caption of figure 2C it reads: Features enriched at baseline and follow-up are represented with negative and positive fold change values respectively. Would it not make more sense to have enrichment in baseline be negative while enrichment in follow up positive fold change values?
Figure 2C caption does state that features enriched at baseline are represented with negative fold change values and features enriched at follow-up are represented with positive fold change values.
8. The figures representing the data of the taxonomic analysis (figures 3 and 4) show taxa on multiple taxonomic levels, although most are on genus level. The authors should either choose a specific level for the analysis or have a separate figure for the levels discussed
We did not perform differential abundance analysis at different taxonomic rank. In each figure, OTUs are indicated at the lowest taxonomic level they were classified based on the SILVA taxonomy vs 138.1. To avoid readers misinterpreting the data, we have provided further clarification in the figure legends by adding the following: “Each OTUs is represented at the lowest taxonomic rank it could be classified accordingly with the SILVA taxonomy (release 138.1).”
9. Figure 5 the bar charts for the follow-up groups are not coloured
We have colour coded the follow-up groups to be the same as baseline to minimise confusion.
10. In the discussion the authors compare their findings to other bariatric surgery studies. Including Roux-Y-Gastric-Bypass studies. Most readers of microorganism will not know the differences in the gut architectures and environment following RYGB and gastric-sleeve. This should be briefly explained which would explain maybe differences in findings between gastric-sleeve and RYGB studies.
We have added a short description of sleeve gastrectomy and RYGB at lines 344 – 348.
Reviewer 2 Report
The article under review is an experimental study of the relationship between the gut microbiome and associated bacterial metabolites. The authors present a cohort study, clearly prescribe inclusion and exclusion criteria for the main group of patients. The authors do not describe the calculation of the allowable sample size, but given the features of the choice of the main group, the sample size is not high. The main advantage of this work is the choice of parameters, both biochemical and metabolic, to search for correlations. Studies of the gut microbiome have been actively developing recently, so obtaining new data in this direction is relevant and timely. It is obvious that the work is of scientific interest, but contains serious remarks. Below is a list of comments and recommendations.
Introduction.
L. 48: Taxonomy - data must be fully corrected in accordance with modern bacterial taxonomy. See: Oren A, Garrity GM. Valid publication of the names of forty-two phyla of prokaryotes. Int J Syst Evol Microbiol. 2021 Oct;71(10). doi: 10.1099/ijsem.0.005056.
L. 58: Meta-analysis – the age about 10 years, try to find more informative, exp. : Huang J, Chen Y, Wang X, Wang C, Yang J, Guan B. Change in Adipokines and Gastrointestinal Hormones After Bariatric Surgery: a Meta-analysis. Obes Surg. 2023 Mar;33(3):789-806. doi: 10.1007/s11695-022-06444-8.
L. 70: Recent reference of 2017 year – try to argument for hypothesis with modern information.
Overall opinion on the Introduction: among 12 references only one dated recent 5-year period 2018-2022; introduction must be improved with recent references and advantages and disadvantages of the main hypothesis.
The article under review is an experimental study of the relationship between the gut microbiome and associated bacterial metabolites. The authors present a cohort study, clearly prescribe inclusion and exclusion criteria for the main group of patients. The authors do not describe the calculation of the allowable sample size, but given the features of the choice of the main group, the sample size is not high. The main advantage of this work is the choice of parameters, both biochemical and metabolic, to search for correlations. Studies of the gut microbiome have been actively developing recently, so obtaining new data in this direction is relevant and timely. It is obvious that the work is of scientific interest, but contains serious remarks. Below is a list of comments and recommendations.
Introduction.
L. 48: Taxonomy - data must be fully corrected in accordance with modern bacterial taxonomy. See: Oren A, Garrity GM. Valid publication of the names of forty-two phyla of prokaryotes. Int J Syst Evol Microbiol. 2021 Oct;71(10). doi: 10.1099/ijsem.0.005056.
L. 58: Meta-analysis – the age about 10 years, try to find more informative, exp. : Huang J, Chen Y, Wang X, Wang C, Yang J, Guan B. Change in Adipokines and Gastrointestinal Hormones After Bariatric Surgery: a Meta-analysis. Obes Surg. 2023 Mar;33(3):789-806. doi: 10.1007/s11695-022-06444-8.
L. 70: Recent reference of 2017 year – try to argument for hypothesis with modern information.
Overall opinion on the Introduction: among 12 references only one dated recent 5-year period 2018-2022; introduction must be improved with recent references and advantages and disadvantages of the main hypothesis.
Methods
L. 179-185: Data on amplicon libraries must be deposit in public database before manuscript submission, I could not check the quality of the data.
Results
L. 200-201: It is necessary to add a table with data on the main group and healthy volunteers to compare them. This table might be either in the main text or as a supplementary file.
L. 206-210: Where are data on alpha diversity indexes on healthy volunteers? No figure, no words…
The authors used only one alpha diversity index for diversity estimation. At least 3 parameters, better 5, would be compared for this analysis. Different indexes reflect different peculiarities of community diversity, and it is correct to compare them.
L. 223-224: The authors wrote that they found “marked differences in the stool taxonomic profiles at both baseline and follow-up visits” for the main group and healthy volunteers, that means the effect of bacteria on the healthy status of patients.
Main remarks on the results section
The authors had to add information on the baseline and follow-up characteristics of healthy volunteers and compare the two groups. The authors had unique data on microbial profile in fecal and microbial metabolites in blood; they introduced only measurement data, not including bioinformation and statistical analysis. The authors could analyze the predicted metabolic function of microbes by PICRUST soft and add data on correlation analysis of microbial profile and metabolites. The manuscript must be supplied with additional results for further discussion.
Discussion
L. 293-297 and below: Add data on predicted metabolic function and correlation analysis to the results.
From L. 353: Discuss more about limitation of the research.
Author Response
Reviewer 2:
1. L. 48: Taxonomy - data must be fully corrected in accordance with modern bacterial taxonomy. See: Oren A, Garrity GM. Valid publication of the names of forty-two phyla of prokaryotes. Int J Syst Evol Microbiol. 2021 Oct;71(10). doi: 10.1099/ijsem.0.005056.
The names of the bacterial clusters have been corrected in accordance with modern bacterial taxonomy (lines 51-54).
2. L. 58: Meta-analysis – the age about 10 years, try to find more informative, exp. : Huang J, Chen Y, Wang X, Wang C, Yang J, Guan B. Change in Adipokines and Gastrointestinal Hormones After Bariatric Surgery: a Meta-analysis. Obes Surg. 2023 Mar;33(3):789-806. doi: 10.1007/s11695-022-06444-8.
Thank you for the suggestion. We have now updated the meta-analysis reference to the suggested, more current reference.
3. L. 70: Recent reference of 2017 year – try to argument for hypothesis with modern information.
Thank you for the suggestion. We have now updated the meta-analysis reference to a more current one.
4. Overall opinion on the Introduction: among 12 references only one dated recent 5-year period 2018-2022; introduction must be improved with recent references and advantages and disadvantages of the main hypothesis.
We have updated the introduction with references from years 2018 – 2023 as suggested by the reviewer and included discussion on the advantages and disadvantages of the hypothesis (lines 78-84).
5. L. 179-185: Data on amplicon libraries must be deposit in public database before manuscript submission, I could not check the quality of the data.
We acknowledge that deposition of the data in a public database facilitates transparency of the analysis. However, we must first seek permission from our relevant ethics committees before doing this. We are happy to do so if the Editor feels this is a requirement for publication.
6. L. 200-201: It is necessary to add a table with data on the main group and healthy volunteers to compare them. This table might be either in the main text or as a supplementary file.
We have now added a table with baseline and follow-up characteristics of the healthy volunteers as Supplementary Table 1. Limited clinical data was available on these people as they were not patients in the hospital.
7. L. 206-210: Where are data on alpha diversity indexes on healthy volunteers? No figure, no words…
Alpha diversity indexes were measured on samples collected from the healthy volunteers and we have now included this as Supplementary Figure 1. No forest plots were generated as there were no significant differences. We have also added text explaining this with reference to the supplementary figure (lines 229-232) in the manuscript.
8. The authors used only one alpha diversity index for diversity estimation. At least 3 parameters, better 5, would be compared for this analysis. Different indexes reflect different peculiarities of community diversity, and it is correct to compare them.
We appreciate the comment made by the reviewer. We agree that different ecological indexes provide different information from the community, but they can also be biased by rare or dominant species, or even differences in sequencing depth between samples. Shannon diversity combines measures of richness (species abundance values in the measurement of biodiversity) and evaluate species heterogeneity, therefore combining measures of richness and abundance. Furthermore, Shannon diversity is equally sensitive to rare and abundant species, and as a compound diversity measure is also characterized for being less dependent on sampling effort, which is a common issue in 16S-based amplicon sequencing studies. Shannon index is also a commonly used diversity metric in the literature. Thus, we consider that this ecological index provides the reader with a good overview of the structure of the gut microbial communities associated with the clinical groups evaluated in our work.
9. L. 223-224: The authors wrote that they found “marked differences in the stool taxonomic profiles at both baseline and follow-up visits” for the main group and healthy volunteers, that means the effect of bacteria on the healthy status of patients.
We apologise for the confusion; we have now clarified that the differences were between bariatric and healthy groups at baseline and bariatric and healthy groups at follow-up timepoints. There were no changes in the healthy group between baseline and follow-up. This has been amended in the abstract (line 41) and in the text (lines 246-251).
10. The authors had to add information on the baseline and follow-up characteristics of healthy volunteers and compare the two groups. The authors had unique data on microbial profile in fecal and microbial metabolites in blood; they introduced only measurement data, not including bioinformation and statistical analysis. The authors could analyze the predicted metabolic function of microbes by PICRUST soft and add data on correlation analysis of microbial profile and metabolites. The manuscript must be supplied with additional results for further discussion.
As outlined in our response to Reviewer 1, Comment 2, this was a pilot study where the objective was to focus on characterising the gut microbiota in patients undergoing bariatric surgery. Predicted functional approaches using tools such as PICRUST have recognized limitations, which make it difficult to make relevant biological interpretations related to microbiota and metabolites. As pointed out by this Reviewer, a strength of our manuscript centres on the fact that the data sets included for microbiota and metabolites are all experimentally generated. Consequently, we opted to rely on experimentally determined robust data sets as opposed to using less reliable predicted data set in attempting to establish biological functions and significance issues. However, we have included additional text on the limitations of the approach in the manuscript, Page 11, Lines 400 – 405.
11. L. 293-297 and below: Add data on predicted metabolic function and correlation analysis to the results.
Please see response to Reviewer 2, Comment 10.
12. From L. 353: Discuss more about limitation of the research.
We have added additional discussion on the limitations (Page 11, Lines 391-405).
Round 2
Reviewer 1 Report
I am satisfied with the answers to my comments and the modification of the manuscript.